# A Highly Sensitive and Flexible Capacitive Pressure Sensor Based on a Porous Three-Dimensional PDMS/Microsphere Composite

**DOI:** 10.3390/polym12061412

**Published:** 2020-06-24

**Authors:** Young Jung, Wookjin Lee, Kyungkuk Jung, Byunggeon Park, Jinhyoung Park, Jongsoo Ko, Hanchul Cho

**Affiliations:** 1Precision Mechanical Process and Control R&D Group, Korea Institute of Industrial Technology, 42-7, Baegyang-daero 804beon-gil, Sasang-gu, Busan 46938, Korea; young89@kitech.re.kr (Y.J.); bgpark91@kitech.re.kr (B.P.); 2Graduate School of Mechanical Engineering, Pusan National University, Busandaehak-ro 63beon-gil, Geumjeong-gu, Busan 46241, Korea; mems@pusan.ac.kr; 3Advanced Surface Coating & Processing R&D Group, Korea Institute of Industrial Technology, 14, Namyangsan 1-gil, Doing-myeon, Gyeongsangnam-do, Yangsan 50635, Korea; wkjinlee@kitech.re.kr; 4Quality&Standards Department, Korea Marine Equipment Research Institute, 435, Haeyang-ro, Yeongdo-gu, Busan 49111, Korea; kkjung@komeri.re.kr; 5School of Mechatronics Engineering, Korea University of Technology & Education, 600, Chungjeol-ro, Byeongcheon-myeon, Dongnam-gu, Cheonan-si, Chungcheongnam-do 31253, Korea; jhpark98@koreatech.ac.kr

**Keywords:** pressure sensors, polymeric composite, microporous, porous structure, microspheres

## Abstract

In recent times, polymer-based flexible pressure sensors have been attracting a lot of attention because of their various applications. A highly sensitive and flexible sensor is suggested, capable of being attached to the human body, based on a three-dimensional dielectric elastomeric structure of polydimethylsiloxane (PDMS) and microsphere composite. This sensor has maximal porosity due to macropores created by sacrificial layer grains and micropores generated by microspheres pre-mixed with PDMS, allowing it to operate at a wider pressure range (~150 kPa) while maintaining a sensitivity (of 0.124 kPa^−1^ in a range of 0~15 kPa) better than in previous studies. The maximized pores can cause deformation in the structure, allowing for the detection of small changes in pressure. In addition to exhibiting a fast rise time (~167 ms) and fall time (~117 ms), as well as excellent reproducibility, the fabricated pressure sensor exhibits reliability in its response to repeated mechanical stimuli (2.5 kPa, 1000 cycles). As an application, we develop a wearable device for monitoring repeated tiny motions, such as the pulse on the human neck and swallowing at the Adam’s apple. This sensory device is also used to detect movements in the index finger and to monitor an insole system in real-time.

## 1. Introduction

Recently, polymer-based flexible pressure sensors have been attracting attention due to their applications in electronic skin [1,2], flexible touch display [3], health care [4,5,6,7], and human–machine interfaces to facilitate human functionalities [8]. The sensors are required to exhibit high sensitivity, low power consumption, and a wide range of operating pressure—from soft touches (i.e., 0–10 kPa considered as the low-pressure range) to object handling (i.e., 10–100 kPa considered as the medium-pressure range). To fulfill these aims, many researchers have reported fabricating a flexible and wearable pressure sensor with high sensitivity by employing various sensing mechanisms such as piezoresistivity [9,10,11,12,13], capacitance [14,15,16,17], piezoelectricity [18,19], and triboelectricity [20,21,22,23]. Among these, capacitive-based sensors have several advantages, including a simple design, stable signals, high reproducibility, and low hysteresis [14]. These sensors perceive the magnitude of pressure by a changed capacitance value that increases or decreases inversely as the distance between the two parallel sensing electrodes changes due to external pressure.

The capacitance value of the capacitive-based sensors depends on the dielectric constant of material (*ε*), the distance between the two electrodes (*d*), and the area between the electrodes (*A*). In particular, changing the distance and dielectric constant can improve device sensitivity. Recent efforts, however, have focused on enhancing the sensor sensitivity based on microstructured polymer dielectric layers that are easily compressible with pressure. Mannsfeld et al. fabricated a pressure sensor with a high sensitivity of 0.55 kPa^−1^ in the low-pressure range (<2 kPa) based on a pyramid shape microstructure [24], and Benjamin et al. proposed a sensor based on the different spatial configuration of microstructured polymer dielectric film to reduce the effective mechanical modulus [25]. Pang et al. reported pressure-sensitive skin-conformal sensors with a high sensitivity (0.53~0.58 kPa^−1^) for detecting radial artery pulse waves based on the microhair-structured biocompatible polymer and micro-pyramid structure [26]. However, the fabrication of the silicon mold pattern is expensive, complex, and requires multi-step processes such as patterning and etching. Additionally, the pressure detection range is limited to 10 kPa, which is insufficient for wearable systems as they require the capacity to detect subtle change (above 10 kPa).

To detect higher pressure ranges with high sensitivity, commercially available polymeric three-dimensional (3-D) porous structures are employed as a dielectric layer in capacitive type sensors. The polymeric porous structures can easily deform even at low pressures, owing to reduced stiffness and the anti-barreling phenomenon [27]. During the unloading state, due to the low viscoelastic property, a porous structure reverses to its original state without any physical defect. These features make sensors that are based on porous structures useful for monitoring human motion. Various methods of fabricating polymer-based porous structures such as 3-D printing [28,29], gas foaming [30], phase separation [31,32], and the templating process [33,34,35,36] have been introduced over the last 10 years. Of these, the templating process offers ease of fabrication, broad applicability, eco-friendliness, and uniformity of the templating process using selectively soluble sugar and salt in water. Research has shown that these pressure sensors have high sensitivity and wide pressure ranges [14,34,36]. Han et al. showed that the carbon nanotube (CNT)-polydimethylsiloxane (PDMS) sponge could be applied to pressure sensors via conductive pathway change to withstand compressive stress [36]. Iglio et al. also suggested PDMS sponges decorated with CNT for detecting subtle pressure (20 Pa) to a larger working range (50 kPa) [37]. Kwon et al. reported a pressure sensor based on the giant piezocapacitive effect of a porous dielectric elastomer having a softer elastomer and lower elasticity than PDMS, and capable of wide pressure detection range (~130 kPa) [27]. These studies focused on enhancing performance by using conductive nanomaterials on the sponge or changing the polymer materials while maintaining the porosity. While the above methods can improve the performance of the sensors, it cannot step up or tune the sensor sensitivity. The easiest and fastest way to tune the sensor sensitivity in stages is to adjust its porosity. As the porosity increases, relative permittivity (piezocapacitive effect) changes along with the mechanical properties, making it possible to improve sensor sensitivity step by step as desired.

In this study, we developed a flexible and highly sensitive pressure sensor based on a microporous structure without altering the material of the elastomer by maximizing porosity using microsphere (MS). The mechanical properties of the porous polymeric structure via the templating process, including compress/release response under static and dynamic compression, were analyzed. The polydimethylsiloxane (PDMS) and MS composite structure (PDMS/MS composite) show improved compressibility and reduced elastic modulus more than the bare PDMS structure because of their increased porosity. We found that as the concentration of MS increases, the sensor sensitivity improves gradually. The sensitivity is found to increase by more than three times for a 10% increase in porosity. Furthermore, finite element (FE) simulations are employed to understand the compression behavior of the porous PDMS/MS composite and PDMS structure as a function of the diameter of the strut and length of the cubit unit cell. The PDMS/MS composite-based sensors shows higher sensitivity than sensors based on PDMS structure, not only at the same pressure but even at the same strain due to the high compressibility and enhanced piezocapacitive effect. Our pressure sensors exhibited excellent outstanding performance of 0.124 kPa^−1^ at 15 kPa and a wide pressure range, specifically between 0 and 140 kPa with high repeatability, reproducibility, and reliability of up to 1000 cycles at 2.5 kPa with negligible hysteresis. We also demonstrated the use of the proposed sensor to detect low (pulse monitoring at the human neck, swallowing at the Adam’s apple) and high (index finger motion, real-time monitoring at the insole system) pressure levels.

## 2. Materials and Methods 

### 2.1. Preparation of the PDMS/MS Composite and Capacitive Pressure Sensors

A PDMS precursor (Sylgard 184, Dow Corning Corp., USA) with a mixture of Resin and Hardener at a 10:1 ratio was prepared. MS (MFL-100MCA, SDI, Korea) was mixed with PDMS precursor and then placed in a vacuum chamber (Jeio Tech, Korea) under 1 × 10^−1^ torr for 10 min to remove the bubbles. A sugar cube was placed onto a Petri dish, and the PDMS and MS mixture were poured into the Petri dish. The empty spaces of the sugar cube mold were filled with the PDMS and MS mixture owing to the capillary effect. After the infiltration of PDMS and MS mixture, the sugar cube was placed in the oven and cured for 5 h at 75 °C. After the curing process, the sugar was dissolved in water, and absorbed water in the porous structure was later dried for 2 h at 100 °C. As the sugar grains were dissolved in water, the empty spaces became pores in the porous structure. The indium-tin-oxide (ITO) coating polyethylene terephthalate (PET) films (Fine chemical Industry, Korea) were used as electrodes and attached to the top and bottom of the sponge using silver paste (P-100, Jin Chem, Korea). The silver paste was thermally cured for 30 min at 100 °C, to remove the remaining solvent. The sensing lines are connected to PET films using silver conductive epoxy (CW-2400, ITW Chemtronics, USA).

### 2.2. Characterization of the PDMS/MS Composite

The morphology of the PDMS/MS composites was characterized via field emission scanning electron microscopy (FE-SEM, S-4800, HITACHI, Japan). For the measurement of compressive stress–strain curves at the various compressive strain ranges, a universal testing machine (JSV-H1000, Japan) was employed to characterize the performance with the load cell (HF-1, maximum load 10 N, load resolution 0.001 N). The composite was compressed at a speed of 10% strain min^−1^ except for the strain rate test in mechanical evaluation. The capacitance of the pressure sensors was measured using an LCR-meter (HIOKI-3536, HIOKI, Japan) with 200 kHz, 1 V bias. All sensor evaluations were measured by connecting the LCR-meter and computer for real-time measuring.

## 3. Results and Discussion

Figure 1a schematically illustrates the fabrication process of the PDMS/MS composite microporous structure via the capillary effect. The images of the surface and the cross-section of the PDMS/MS composite obtained through FE-SEM are given in Figure 1b. The MS, which has a specific gravity of about 0.12, consisting of thermoplastic resin and hydrocarbon (Appendix A) inside the PDMS support structure, contributes to maximizing the porosity of the sponge as the micro-size pores. MS are randomly and densely distributed inside the PDMS structure layer, becoming the micropores inside the support column and layer of the PDMS structure, as shown in Figure 1b and Appendix A. On the other hand, the bare porous PDMS structure only consists of a PDMS support structure with a smooth surface, as shown in Appendix A. The average sizes of macro and micropores of the PDMS/MS composite were measured to be 523 ± 78 μm and 55.7 ± 8.5 μm, respectively. The sizes of the macropores are a good match with the grain size of the sugar cube. Figure 1c shows the photograph of the fabricated 3-D porous PDMS/MS composite structure and the mechanical robustness under the various deformation modes such as compressing, stretching, bending, and twisting.

Figure 2 illustrates the working principle of the 3-D porous PDMS structure and the proposed porosity-maximized PDMS/MS composite structure. The 3-D porous PDMS structure has porosity in proportion to that of the sugar particles, while the 3-D porous PDMS/MS composite structure has a higher porosity because the ratio of the MS to the pores contributes to the ratio of the sugar particles. Because of the reduced stiffness, the composite structure exhibits high deformation even at the same pressure resulting in high sensitivity. The sensitivity of the capacitive type pressure sensors is determined by the distance between the two parallel sensing electrodes. The dielectric layer based on a porous structure is easily compressed (from *d* to *d*′) by pressure, and the sensitivity of the sensor becomes high. Furthermore, for porous structures, the relative permittivity increases (from *ε* to *ε*′) when external pressure is applied. We conducted a theoretical analysis to determine the effect of the distance between electrodes and relative permittivity on sensor sensitivity. The capacitance of the parallel-plate capacitor is related by
(1)$$C=\varepsilon_{0}\varepsilon_{r}\frac{A}{d}$$
where *ε*_0_ is the vacuum permittivity, *ε_r_* is the relative permittivity, *A* is the area of the electrode, and d is the distance between the two parallel sensing electrodes. The relative change in capacitance is calculated as follows:(2)$$\frac{\Delta C}{C_{0}}=\frac{C-C_{0}}{C_{0}}=\frac{C}{C_{0}}-1$$

Substituting the initial and compression state capacitance in Equation (2), we get
(3)$$\frac{\Delta C}{C_{0}}=\frac{d_{0}{\varepsilon_{r}}^{\prime }}{d_{c}\varepsilon_{r}}-1=\frac{d_{0}}{d_{c}}\frac{f\varepsilon_{a}+\left(1-f\right)\varepsilon_{p}}{f_{0}\varepsilon_{a}+\left(1-f_{0}\right)\varepsilon_{p}}-1$$
where the terms *d*_0_ and *d_c_* correspond to the initial and compressed distance between the two electrodes, *ε_a_* and *ε_p_* are the dielectric constants of air and PDMS respectively, and the initial and compressed air fraction are *f*_0_ and *f*, respectively. The relative permittivity can be expressed as the product of air fraction (*f*, porosity) and the dielectric constant (*ε_a_, ε_b_*).

We assumed the compressed air fraction *f* decreased linearly with the compressive strain *ϵ*, and substituting for *ε_a_* = 1, *ε_p_* = 3 in Equation (3), we get
(4)$$\frac{\Delta C}{C_{0}}=\frac{d_{0}}{d_{c}}\left(1-\frac{2a\epsilon }{2f_{0}-3}\right)-1$$

Details about the theoretical analysis and the modeling assumptions are found in the “Appendix A” section. From Equation (4) we can see that the ratio of initial and compression distance between the two electrodes is more important to improve the sensitivity. The PDMS/MS composite with high porosity is significantly compressed compared to the PDMS structure at the identical pressure because of its reduced stiffness. Therefore, high sensitivity can be achieved by using a microporous structure with maximum porosity.

To verify the porosity enhancement of the PDMS/MS composite, we calculated the porosity of the porous structure according to Appendix A. We measured the weight of the bulk PDMS and PDMS/MS composite, porous PDMS, and PDMS/MS composite. The volume of all the specimens was equal to 16 × 16 × 10 cm^3^. As shown in Appendix A, the porosity of the PDMS structure was measured to be 65.3%, whereas the porosity of the PDMS/MS composite was 75.6% in 9 wt.% of MS, which is higher than that of the PDMS structure. These results show that the MS maximizes the porosity by acting as a micropore in the porous structure.

Figure 3a shows images of the fabricated PDMS/MS composite when it is compressed at 60% strain to evaluate its mechanical properties. The mechanical properties of the PDMS/MS composite are different because of the change in porosity resulting from the concentration of the MS. We quantitatively measured the compressive stress–strain of the PDMS/MS composite at 60% strain at different MS concentrations (Figure 3b). As the MS concentration increased, compressive stress decreased until it reached 9 wt.%. As the MS concentration increased from 9 wt.% to 11 wt.%, the maximum compressive stress decreased by 0.4%. The results indicate that the 9 wt.% concentration of the MS is the fabrication limit via the capillary effect because of the high viscosity. The 9 wt.% of PDMS/MS composite was therefore used in the measurements of mechanical properties and capacitive pressure sensors. To compare the PDMS/MS composite and bare PDMS structure, as shown in Figure 3c, the compressive stress–strain of the porous structure was measured with up to 60% compression with 10% strain min^−1^. In cases of the PDMS structure, a pressure of about 42.97 kPa was required for 60% strain, while the PDMS/MS composite required pressure of only 14.07 kPa for the same strain. The results represent that the addition of MS into the porous structure reduces the stiffness and makes the structure easily compressible with pressure. As MS has low specific gravity, it can contribute to aggregation and non-uniform arrangement on the porous structure. For investigating the uniformity of the porous structure, we evaluated mechanical properties with compression direction in the x, y, z-directions (Figure 3d). We verified that the error of compressive stress–strain in each direction was within 1%, which means that the PDMS/MS composite exhibits isotropic sensing performance and can be used in any direction. Furthermore, we measured the compression stress at 20%, 40%, and 60% strain, as shown in Figure 3e. The compressive stress rapidly increased with an increase in strain. At maximum strain, the area between the loading and unloading curves (hysteresis) was bigger. Usually, when pressure is applied to a polymer, the polymer absorbs the potential energy and uses it to recover to its original state [38]. The area between the loading and unloading curves represents the energy loss in the compression process, which is why hysteresis occurs. In our experiments, the loading and unloading curves almost overlapped, thereby ensuring minimal hysteresis in the sensors. Finally, we assessed the compressive stress–strain when the PDMS/MS composite was compressed at different strain rates (10~1500% strain min^−1^) for loading and unloading (Figure 3f). High compressive stress of up to 500% strain min^−1^ is required to deform the porous structure at a faster compression rate. The measurement results exhibited negligible differences and high stability characteristics over the entire strain rate, making it useful for applications that require detecting various frequencies without additional calibration. 

Finite element (FE) simulations of the compressive tests were conducted using a porous lattice unit cell structure to understand the compression behavior of the PDMS and PDMS/MS composites. As shown in Figure 4a, the porous structure was idealized by a representative volume element consisting of 27 unit cells. A body-centered cubic strut structure was used for each unit cell. The simulations employed the orthogonal mixed boundary conditions [39], i.e., the boundary walls remained flat during deformation. Experimental compressive stress–strain curves of the bulk PDMS and PDMS/MS composite were used to describe the flow rules of the materials in the simulations (Appendix A).

The simulations were conducted with varying *d/L* of the unit cell, where *d* is a diameter of the strut, and *L* is the length of the cubic unit cell (Appendix A). It was found that the FE model with *d/L* = 0.2575 had a similar compressive curve to the porous PDMS structure. Figure 4a shows the evolution of the internal equivalent stress of the porous PDMS structure, calculated by the FE model with *d/L* = 0.2575. The resulting stress–strain curve was compared to the experimental result in Figure 4b. The shape of the compressive curve from the FE simulation is slightly different from the experiment, probably due to the fact that all the strut walls contact each other at the same compressive strain in the FE simulation by the idealized microstructure, whereas the shape of the pores is irregular in the real microstructure of the porous PDMS structure. For the PDMS/MS composite, a good correlation can be observed between the experiment and the FE simulation when *d/L* = 0.245, which is only very slightly lower than the value obtained for the porous PDMS structure (shown in Figure 4c), indicating that the PDMS structure and PDMS/MS composites have similar pore morphology and micro-architecture. 

The FE model used for the PDMS (*d/L* = 0.2575) in Figure 4b has a porosity of 72.3%, which is comparably higher than the actual porosity of the porous PDMS structure of 65.3%. For PDMS/MS composite, the porosity of the model was 74.6% (when *d/L* = 0.245), excluding the porosity due to the internal closed pores introduced by the MS. A relatively poor internal connectivity in the microstructure of the porous structure due to the irregular pore morphology could be a possible reason a higher porosity is required in the idealized FE model to reproduce the experimentally observed compressive behavior. In the FE simulations, the applied load is borne by the whole microstructure because of the idealized lattice structure, whereas it is expected that some internal parts of the real sponge could not contribute to the strength due to less connectivity. 

The internal stress during compression is significantly higher than the applied compressive stress in both materials due to the porosities, as shown in Figure 4a. When comparing the evolutions of the equivalent stress while in compression between the PDMS structure and PDMS/MS composite, significantly lower stress was observed in the PDMS/MS composite than in the PDMS structure. When the applied compressive strain was 60%, the applied overall compressive stresses were ~40 kPa and ~16 kPa for the PDMS structure and PDMS/MS composite models, respectively. The internal stress of the PDMS structural model at the same applied strain was in the range of 250~300 kPa, compared to the PDMS/MS composite model with a stress range between 100~150 kPa due to the improved flexibility caused by the addition of the MS, which has internal pores and has significantly lower apparent elastic stiffness than the PDMS structure. 

The PDMS/MS composite with high porosity was used as the dielectric layer of the capacitive pressure sensor. According to Equation (4), high compressibility (*d*_0_*/d_c_*) and high initial porosity (*f*_0_) are important factors that increase the sensitivity of the sensors. At identical pressures, the sensor based on the PDMS/MS composite can be easily compressed by its reduced stiffness compared to the PDMS sensor. High compressibility changes the distance between the two electrodes and increases the capacitance values. Figure 5a shows the results of the measurement of capacitance variation with the same pressure (1.5 kPa) loading and unloading four cycles to the sensor based on the PDMS/MS composite and PDMS structure according to time. This shows that when porosity is increased by 10%, the sensitivity rises by roughly 4 times at 1.5 kPa. Even at the identical strain, high initial porosity can enhance the sensitivity because of the effective relative permittivity changes. Figure 5b shows the measurements of capacitance values of the PDMS/MS sensor and PDMS sensor with the same strain (0~60%). Below 20% strain, the capacitance values of the two sensors were almost equal. However, after 20% strain, the difference increased, and the PDMS/MS sensor shows 1.08 times higher sensitivity even at 60% strain. These values are in good agreement with analytical calculations, as shown in the Figure 5b inset. Details about the analytical calculations are found in the Appendix A. These results are representative of why we have focused on porosity. Additionally, a stable response was observed even with repetitive pressure and strain applied to the sensor.

The sensitivity of the PDMS/MS sensor, which is a vital factor for practical applications, was evaluated under applied static and dynamic pressure. Figure 5c shows measured capacitance changes (Δ*C/C*_0_) as a function of the pressure of the PDMS/MS sensors. The sensitivity is defined as *S = δ(*Δ*C/C*_0_*)/δp*, where *C* and *C*_0_ denote the capacitances with and without applied pressure, respectively, and *p* denotes the applied pressure. The pressure sensitivity of the PDMS/MS sensor is divided into three linear sections. From 0 to 15 kPa, sensitivity is highest, with the sensitivity of the PDMS/MS sensor being 0.124 kPa^−1^ in this section, which is 3 times higher than that of the PDMS sensor. The sensitivity in the second linear pressure section (15~55 kPa) is also 1.5 times higher than the maximum sensitivity of the PDMS sensor. In the third linear region, which exhibits the lowest sensitivity, the pores of the two sensors are completely closed, acting as solid PDMS, and the sensitivities of the two sensors are very similar. The reason for the enhanced pressure sensitivity of the PDMS/MS composite-based sensor can be two-fold. The first is the maximized porosity of the PDMS/MS composite. With maximized porosity of the porous structure, the deformation amount due to the pressure becomes larger because of reduced stiffness. Therefore, when compression of the PDMS/MS composite occurs more than the PDMS structure, the distance between the electrodes is considerably changed and sensitivity improved. The second reason is the increased dielectric constant with the addition of MS. Some researchers demonstrated that the sensitivity of the sensors with polymer-based microstructure film and the porous structure could be improved by maximizing the piezocapacitive effect [24,25,27,36]. In the case of PDMS/MS composite, it consists of PDMS and a large amount of MS, which consists of thermoplastic resin and hydrocarbon. The dielectric constant of PDMS is as low as 2.7~3.0, but the hydrocarbon is about 30 [40], which is very high. Due to the increased dielectric constant, the sensitivity of the sensor is improved. We further verified the sensing performance for pressure sensors with different thicknesses and MS concentration of porous structure. The PDMS/MS sensor showed a similar sensitivity in the 10 kPa pressure range against the applied pressure (Appendix A). However, at pressure above 10 kPa, the sensitivity of the sensors at 4 and 7 mm decreased due to compression of the porous structure. Additionally, the sensor made of 9 wt.% and with the highest MS concentration yielded the highest sensitivity (0.124 kPa), compared with sensors made of 5 wt.% and 1 wt.%, which yielded sensitivities of 0.099 and 0.069 kPa, respectively (Appendix A).

Figure 5d shows the measured relative changes of capacitance in the pressure sensor based on the PDMS/MS composite with loading and unloading on time in 10% strain min^−1^. The sensors show a stable signal at various pressures (2.5, 7, 16 kPa). To evaluate the reproducibility of the sensors, 0.8, 3.2, and 6.4 kPa pressure was applied to the sensor, and the relative change in capacitance was measured (Figure 5e). No deviation of the capacitance occurred with and without pressure. To calculate the hysteresis error, we measured the capacitance value when the strain was continuously changed (Figure 5f). The measured hysteresis error was less than 0.1% in all steps. We also verified the response time of the sensors measured in 3 kPa (Appendix A). The 10% to 90% rise time (*t_r_*) and 90% to 10% fall time (*t_d_*) were measured to be 167 ms and 117 ms respectively. The sensors based on the PDMS/MS composite consists of PDMS support structure, which has excellent elastic properties. The sensors are immediately compressed with pressure and released without pressure. Even when pressure is immediately applied and removed, the PDMS/MS composite returned quickly to its original shape without any permanent deformation.

Figure 6 shows the measurement results of the reliability test. A pressure of 2.5 kPa was applied to the sensors 1000 times repeatedly while measuring the capacitance values. The initial 10 cycles, the middle 10 cycles, and the last 10 cycles are shown in the red, green, and blue graph, respectively. The measured average capacitance value and deviation of the sensors were 0.353, 1.31% in the initial cycles, 0.346, 1.22% in the middle cycles, and 0.341, 1.45% in the final 10 cycles. Although the measured capacitance values slightly decreased, the standard deviations in all cycles were within 2%. The reason for the high reliability of the fabricated sensor is the high elasticity of PDMS and the buffering effect of MS. When pressure is applied, PDMS absorbs elastic energy, and when the pressure is removed, elastic energy is used to recover to the original state. At this time, the numerous MS inside the sponge function to buffer the working pressure. Therefore, even if a high pressure is applied to the sponge, it can be used as a sensor having high reliability due to the buffering effect. The representative reports on capacitive pressure sensor based on porous PDMS structure with nanoparticles are summarized in Appendix A.

Because of their excellent sensitivity, durability, fast response time, and wide detection range, the fabricated flexible pressure sensors with a PDMS/MS composite could be applied to health monitoring and wearable devices. First, we evaluated the capabilities of these sensors in health monitoring (small deformation). The pressure sensors were tightly attached to the surface of the skin, such as the human neck, through adhesive bandages. As shown in Figure 7a, our sensors could accurately monitor the blood pressure of a human carotid in real time. The neck pulses could be read out clearly under normal conditions (60~66 beats per minute) with regular wave shapes. Under these conditions, a typical carotid pulse waveform with three clear peaks was obtained: the Percussion wave (P-wave), Tidal wave (T-wave), and Diastolic wave (D-wave), which are related to the systolic and diastolic blood pressure, the late systolic augmentation, and the heart rate, respectively [26]. Additionally, we measured motion near the throat when a test subject swallowed their saliva. This motion released the compressed sponge, thus increasing relative capacitance values. Figure 7b shows the sensor with a PDMS/MS composite attached to the Adam’s apple. These measured capacitance values enable us to detect and distinguish the various motions near the throat and can be used to monitor human physiological activities (Figure 7c).

To demonstrate the applicability of our sensor for detecting high pressures, we applied the sensor to a wearable application where the bending angle of the index finger was evaluated. The flexible sensor was placed onto the index finger using a commercial bandage, and its angle was gradually changed as shown in Figure 8a. The angle variations were divided into five stages, starting at a relaxed stage of 0° and then bent at angles of 20°, 40°, 60°, 80°, and 0° respectively. Figure 8b shows the measurement results of the relative capacitance values according to the index finger’s angle. The corresponding relative capacitance values increased significantly with the finger bending angle. Finally, we fabricated a shoe insole with integrated, flexible pressure sensors, and capacitance values from different motions such as running and walking were measured for 10 s in alternation. Figure 8c shows photos of the shoe and shoe insole with attached sensor. It can be observed that the sensor shows different relative capacitance values to different motions (Figure 8d). These results confirm the sensor based on the PDMS/MS composite can accurately sense and distinguish a different scale of pressure.

## 4. Conclusions

We presented a new and facile method for maximizing the porosity of porous structures that could tune mechanical properties via a sugar sacrificial process and MS. The porous PDMS/MS composite, composed of macropores by sugar grain and micropores by MS, can be easily compressed by applying a low pressure because of the increased porosity. The proposed capacitive pressure sensor based on a PDMS/MS composite showed higher sensitivity than the bare PDMS sensor under reduced stiffness and effective relative permittivity at the same pressure and strain. For only a 10% increase in porosity, the proposed PDMS/MS sensor exhibited excellent sensing performance with high sensitivity of 0.124 kPa^−1^ in the pressure range of 0 to 15 kPa as well as stability, reproducibility, low hysteresis, and high durability. To demonstrate the applicability of this pressure sensor on wearable systems, we demonstrated that our flexible sensor can be applied to low- and high-pressure detection. By attaching the PDMS/MS composite sensor to the neck, we showed that tiny variations, such as the movements of the neck’s pulse and the contraction of the Adam’s apple when swallowing saliva, can be measured in real time. Furthermore, we found that this pressure sensor could detect high pressures such as changes in the bending angle of the index finger and different movements in insole systems.

## Figures and Tables

**Figure 1 polymers-12-01412-f001:**
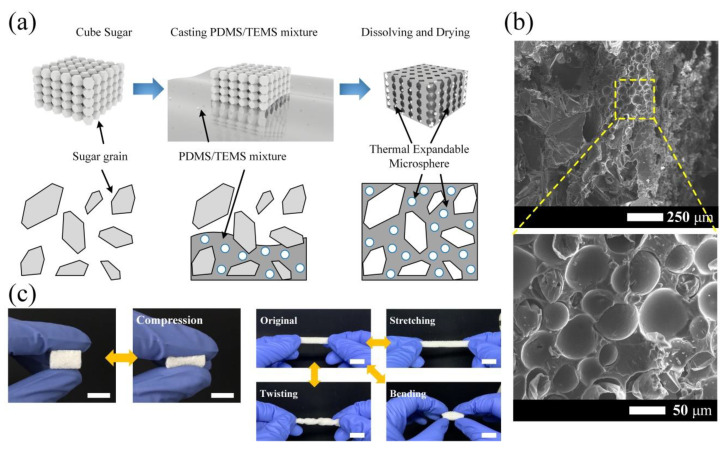
Fabrication process and photos of the PDMS/MS composite. (**a**) Schematic of the PDMS/MS composite using the sugar templating process with precursor solution. The empty spaces of the sugar cube are filled with the PDMS/MS precursor owing to capillary effect. (**b**) FE-SEM images of 3-D porous structure of the PDMS/MS composite. MS are arranged and exposed on the surface of the PDMS/MS composite structure. (**c**) Photographs of fabricated 3-D porous structure and mechanical deformation under various modes; compression, stretching, twisting and bending. (Scale bar: 1 cm).

**Figure 2 polymers-12-01412-f002:**
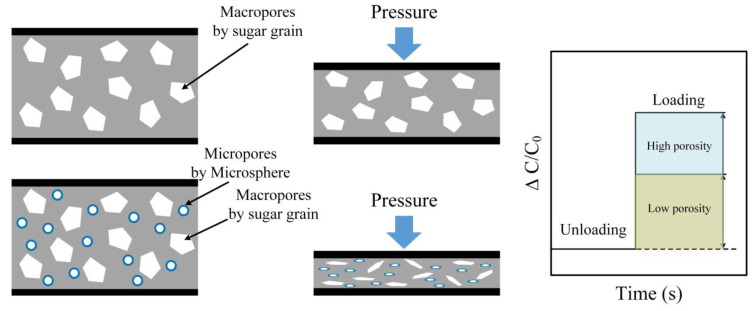
Comparison of the porous PDMS structure and PDMS/MS composite when it compresses with pressure. Bare PDMS structure with low porosity has high elastic resistance, and low sensitivity. However, in cases of the PDMS/MS composite, which has macro and micro pores in the structure, it has low elastic resistance and high sensitivity.

**Figure 3 polymers-12-01412-f003:**
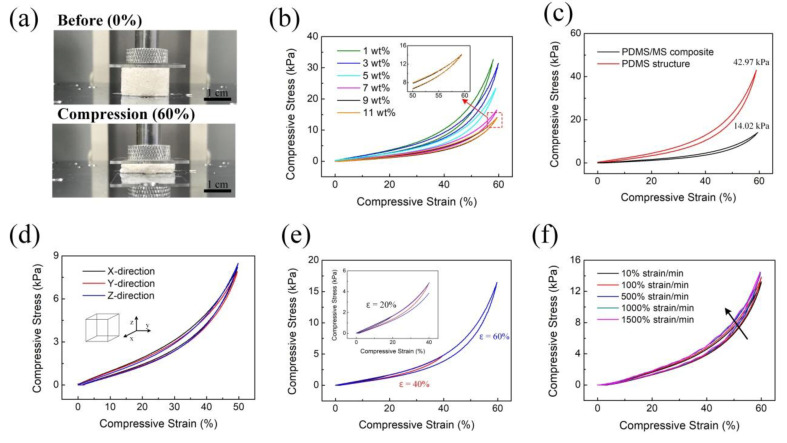
Mechanical evaluation of the PDMS/MS composite. (**a**) The compressive schematic of PDMS/MS composite at 60% strain. (**b**) The measured compressive stress–strain curves according to the concentration of PDMS/MS composite. The curves of 9 and 11 wt.% almost overlap; hence, the 9 wt.% concentration is selected as the maximization point. (**c**) Comparison of compression characteristics of bare PDMS structure and PDMS/MS composite at 60% strain. (**d**) The evaluated compressive stress–strain curves with change in compression direction (x, y, z-direction) for investigating the uniformity of MS. (**e**) The stress–strain curves of PDMS/MS composite with different set strains. (**f**) Stress–strain curves tested under different strain rates.

**Figure 4 polymers-12-01412-f004:**
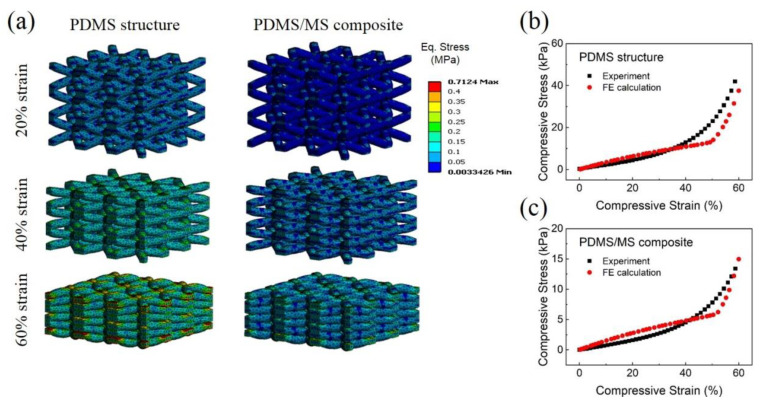
FE simulation for comparison of the compressive behavior of porous PDMS structure and PDMS/MS composite. (**a**) Evaluation of equivalent compressive stress of PDMS structure vs PDMS/MS composite. (**b**) Compressive stress–strain curves of the PDMS structure by experiment and FE calculation. (**c**) Compressive stress–strain curves of the PDMS/MS composite by experiment and FE calculation.

**Figure 5 polymers-12-01412-f005:**
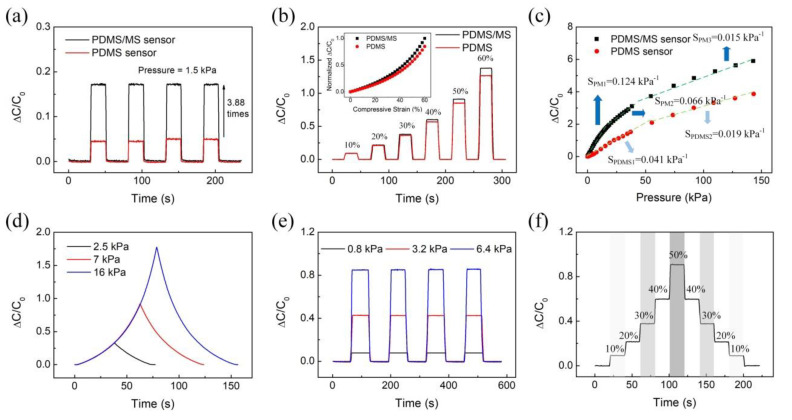
The measured sensing performance of the capacitive pressure sensor based on the PDMS/MS composite. (**a**) Comparison of measurement capacitance variation with loading and unloading with time. (**b**) The capacitance values of PDMS/MS and PDMS sensor with different strain (10~60%). Inset graph shows the analytical calculations. The experimental and analytical calculations show good agreement. (**c**) Pressure-response curves for sensors based on the PDMS/MS composite and bare PDMS structure. (**d**) Variation of capacitance of the PDMS/MS composite with same strain rate (10% strain min^−1^) under different applied stress. (**e**) Reproducibility test of sponge under different applied stress with time. (**f**) The measured hysteresis error in consequent loading and unloading steps.

**Figure 6 polymers-12-01412-f006:**
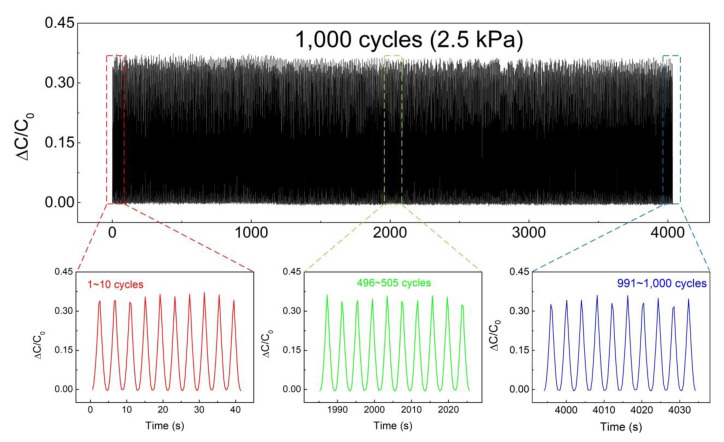
Reliability test results of the pressure sensor over 1000 cycles with measurement capacitance values. The red, green, and blue graphs show measurement results of the initial, middle, and final 10 cycles, respectively.

**Figure 7 polymers-12-01412-f007:**
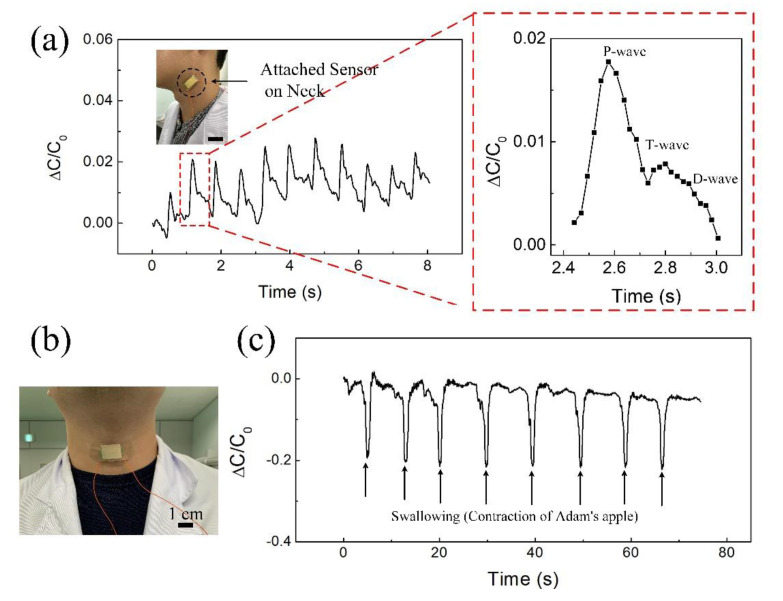
Applications of the pressure sensor to health monitoring. (**a**) Capacitance response of the pressure sensor on the neck. (**b**) Photos of sensor attached to the Adam’s apple. (**c**) Sensing performance of the sensor attached on Adam’s apple by contraction.

**Figure 8 polymers-12-01412-f008:**
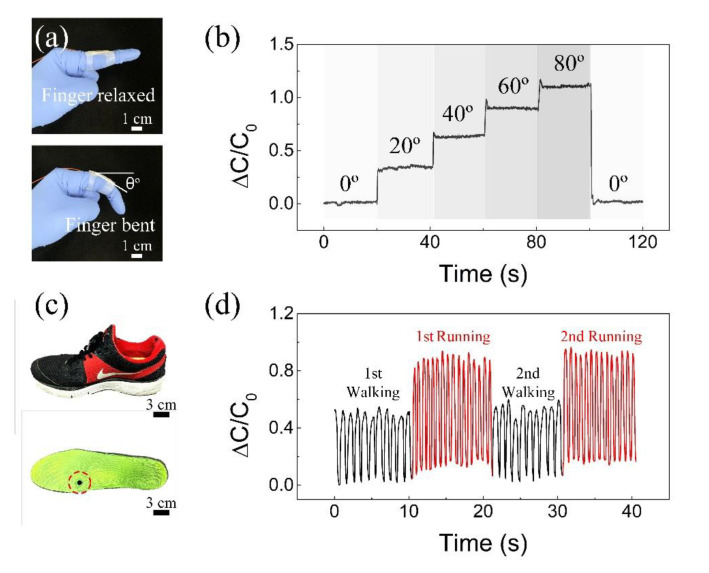
Demonstration of wearable devices with the flexible pressure sensor based on the PDMS/MS composite. For a wearable application, the sensor is placed onto the index finger and sole of the shoe. (**a**) The angle of the index finger is relaxed and bent at various bending stages. (**b**) Measurement results of relative capacitance values according to index finger’s angle. (**c**) Photos of the shoe and shoe insole with an attached sensor. (**d**) Capacitance response of the pressure sensor with repeated motions such as walking and running.

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
