# Peer review of "A Highly Sensitive and Flexible Capacitive Pressure Sensor Based on a Porous Three-Dimensional PDMS/Microsphere Composite"

_polymers, 2020, doi:10.3390/polym12061412_

Round 1

Reviewer 1 Report

The manuscript ‘A Highly Sensitive and Flexible Capacitive Pressure Sensor Based on a Porous Three-Dimensional PDMS/Microsphere Composite’ is mainly focus on fabrication of flexible highly sensing pressure sensor. Authors carried out detailed analysis of the sensing performances. After considering following comments, the work could be accepted in this journal.

Comments#

  1. Authors mentioned that ‘A sugar cube was placed onto a Petri dish, and the PDMS and MS mixture were poured into the Petri dish’ could you please describe more about the properties of MS and in which ratio you mixed PDMS and MS.
  2. Did you observe any diffusion of Ag paste into the sponge during bonding and in which temperature you cured this layers?
  3. Could you please mention the thickness of the sponge and did you observe influence of thickness on the sensing performances?
  4. There are many reports on pressure-sensitive PDMS sponge based on decoration with nanoparticles. Could you please provide a comparison table of the performances of the sensors along with your sensor?
  5. Authors could you please explain (with figures) how the pore size influence on the sensing performances.
  6. Figure of merit related sensing performances is missing in the abstract.

Author Response

We thank the referees and editor for their comments to strengthen the presentation of our results. We have modified the text to respond to all the issues and have elaborated on the changes below.

Reviewer #1

Point 1: Authors mentioned that ‘A sugar cube was placed onto a Petri dish, and the PDMS and MS mixture were poured into the Petri dish’ could you please describe more about the properties of MS and in which ratio you mixed PDMS and MS.

Response 1: Thank you for your valuable comments. We described the properties of MS such as specific gravity and average diameter in the manuscript (page 3, lines 132~133, 139). In order to evaluate properties according to MS concentration, we fabricated PDMS/MS composite with different MS concentration (1~ 11 wt%) and evaluated the mechanical properties. The 9 wt% concentration of the MS showed the valuable mechanical properties and used in the measurements of pressure sensors (page 5, lines 193~200).

Point 2: Did you observe any diffusion of Ag paste into the sponge during bonding and in which temperature you cured this layers?

Response 2: We did not observe the diffusion of Ag paste in this experiment. And we cured Ag paste in the oven for 30 min at 100 °C and modified the manuscript (page 3, lines 117~118).

Point 3: Could you please mention the thickness of the sponge and did you observe influence of thickness on the sensing performances?

→ First, we thank for your comments to strengthen the presentation of our results. We further measured sensing performances according to the thicknesses of the porous PDMS/MS composite. The thicknesses of PDMS/MS composite were controlled to 4, 7, and 10 mm for evaluation. The three sensors showed similar sensitivity in the 10 kPa pressure range against the applied pressure. However, at pressure above 10 kPa, the sensitivity of sensors based at 4 and 7 mm was decreased due to compression of the porous structure. It means that the thickness of the porous structure influences the pressure range (page 8, lines 318~325).

Point 4: There are many reports on pressure-sensitive PDMS sponge based on decoration with nanoparticles. Could you please provide a comparison table of the performances of the sensors along with your sensor?

Response 4: First of all, the point of this paper is that the porosity can be quantitatively controlled using a microsphere and the performance of the sensor can be tuned through this porous structure.

We found the some reports on capacitive pressure sensor based on porous PDMS structure with nanoparticles. And we summarized the specific sensing performances such as sensitivity, pressure range, response time, and reliability for each pressure sensor (page 10, lines 362~363, Table S2).

Point 5: Authors could you please explain (with figures) how the pore size influence on the sensing performances.

Response 5: We also had consideration of pore size and porosity to significantly affect the sensor performance. According to Reference paper (Chen Huang et al., Journal of Applied Mechanics, 2018), porosity, rather than pore size, significantly affects the elastic modulus. We conducted experiments with a focus on increasing porosity and confirmed that the performance is improved compared to the bare porous structure. 

Point 6: Figure of merit related sensing performances is missing in the abstract.

Response 6: We modified the abstract to add the specific evaluation results of the sensor such as sensitivity, response time, pressure detection range, and reliability into the abstract and modified overall corrections (page 1, lines 25~29).

Reviewer 2 Report

The authors present a highly sensitive and flexible pressure sensor using PDMS/microsphere composite that can be used in wearable electronics. The manuscript is well written with solid experimental design and conclusion. The reviewer suggest a minor revision before the manuscript being accepted by Polymers.

Please address the following questions.

1) As stated in the manuscript, porosity is important to the sensitivity of the pressure sensor. Can the authors include data with different porosity rate and show the performance of the sensor?

2) What is the equation of calculating sensitivity?

3) Figure 8a and c both include scale bars with the caption of 'scale bar 1cm'. Please make sure the scale bar reflects the objective you refer to.

Author Response

We thank the referees and editor for their comments to strengthen the presentation of our results. We have modified the text to respond to all the issues and have elaborated on the changes below.

Reviewer #2

The authors present a highly sensitive and flexible pressure sensor using PDMS/microsphere composite that can be used in wearable electronics. The manuscript is well written with solid experimental design and conclusion. The reviewer suggest a minor revision before the manuscript being accepted by Polymers.

Please address the following questions.

Point 1: As stated in the manuscript, porosity is important to the sensitivity of the pressure sensor. Can the authors include data with different porosity rate and show the performance of the sensor?

Response 1: Thank for your valuable comments. We further conducted measurement of sensitivity of sensor with different porosity (microsphere concentration). We confirmed that the sensor based on 1 wt% showed a sensitivity of 0.069, 5 wt% of 0.099, and 9 wt% of 0.124 kPa-1 in a range of 0~15 kPa. It means that the sensitivity of sensor increased as the porosity increased (page 8, lines 316~323).

Point 2: What is the equation of calculating sensitivity?

Response 2: The sensitivity is defined as S =δ(ΔC/C0)/δp, where C and C0 denote the capacitances with and without applied pressure, and p denotes the applied pressure, respectively. We have further described the above the equation of calculating sensitivity on the manuscript (page 8, lines 299~301).

Point 3: Figure 8a and c both include scale bars with the caption of 'scale bar 1cm'. Please make sure the scale bar reflects the objective you refer to.

Response 3: We revised the scale bar in Figure 8a and c as you recommended. And we also make sure the scale bar (pages 11, Figure 7b, Figure 8a and c).

Round 2

Reviewer 1 Report

Authors address all major comments.